# The Effect of SCMs in Blended Cements on Sorption Characteristics of Superabsorbent Polymers

**DOI:** 10.3390/ma14071609

**Published:** 2021-03-25

**Authors:** Rohollah Rostami, Agnieszka J. Klemm, Fernando C. R. Almeida

**Affiliations:** 1School of Computing, Engineering and Built Environment, Glasgow Caledonian University, 70 Cowcaddens Road, Glasgow G4 0BA, UK; A.Klemm@gcu.ac.uk; 2Department of Materials Engineering and Construction, Federal University of Minas Gerais, 31270-901 Belo Horizonte, Brazil; Fernando@demc.ufmg.br

**Keywords:** superabsorbent polymers (SAPs), supplementary cementitious materials (SCMs), characterisation, water absorption capacities (WAC)

## Abstract

Supplementary cementitious materials (SCMs), such as fly ash (FA) and ground granulated blast-furnace slag (GGBS), are often used as a partial replacement of cements to improve the sustainability of Portland cement-based materials and reduce their environmental impact. Superabsorbent polymers (SAPs) can be successfully used as internal curing agents in ultra-high performance cementitious materials by facilitating the hydration process and controlling the water supply in both fresh and hardened states. This paper intends to characterise the physical and chemical properties of SAPs and their sorption properties in different blended cement environments. The swelling capacity and kinetics of absorption of three superabsorbent polymers with different chemical compositions and grading were tested in different cement environments. Experimental results of their sorption performance in distinct solutions, including deionised water (DI), Portland cement (PC), and blended cements (PC-FA and PC-GGBS) and changes in pH of different solutions over time were investigated. The results showed that PC-FA solution had the lowest pH followed by PC-GGBS solution. Moreover, SAPs samples displayed the highest absorption capacities in PC-FA solutions, and the lowest swelling capacities were found in PC-GGBS solutions. Furthermore, SAP with smaller particle sizes had the greatest absorption capacity values in all solutions.

## 1. Introduction

Supplementary cementitious materials (SCMs) have been successfully used for the last couple of decades to improve the sustainability of Portland cement-based materials and reduce their environmental impact. Using SCMs in concrete in blended cements or adding them separately to the concrete mixer is a commonly accepted method that became popular worldwide [1,2,3,4]. Fly ash (FA) from coal combustion (pozzolanic material) and ground granulated blast-furnace slag (GGBS) from pig iron production (latent-hydraulic material) are well-known SCMs, and both are a viable solution to partially substitute Portland cement (PC) [1]. The use of by-products contributes to cost reduction, which is considered as “avoided waste” [2,5] and to the enhancement of concrete properties in the fresh and hardened states [6,7,8,9,10,11]. These improvements can be related to workability [11], lower heat of hydration [11,12,13], better binding capacity [14,15,16], and higher resistance to chloride [17,18] and sulphate attack [19,20].

Nevertheless, the interaction between SCMs and Portland cement during hydration leads to the formation of a complex system [1]. The reactions of pozzolanic SCMs (e.g., FA) with calcium hydroxide (CH) at ambient temperature result in the formation of hydration products such as calcium silicate hydrates (C–S–H) [21]. In case of the latent-hydraulic materials (e.g., GGBS), C-S-H is formed directly by hydration reactions when properly activated [6]. Generally, the pore solution of blended cements with FA and GGBS have lower sodium and potassium concentrations compared to ordinary PC, due to lower alkali concentrations, especially after long hydration times. This reduction is caused by the pozzolanic reactions (formation of C–S–H with a lower Ca/Si ratio and C–A–S–H) and the subsequent decrease of hydroxide concentrations at longer hydration times [22]. Hydroxide concentrations are also lower in the GGBS system [23]. Moreover, pH can be decreased in pore solutions of blended cements, at early hydration times, due to lower alkali content [24,25]. Additionally, it has been widely documented that the pores refinement of cementitious matrices by SCM leads to higher susceptibility to autogenous shrinkage [1,26,27,28,29].

In order to mitigate this negative effect, superabsorbent polymers (SAPs) can be used as internal curing agents and facilitate the hydration process by controlling water supply in both fresh and hardened state [30,31,32,33,34,35]. This application has drawn particular interest in ultra-high-performance cementitious materials [36,37,38,39,40,41,42,43,44,45], which is mainly due to the very low w/c ratios used (below 0.20). SAP can facilitate the hydration process and control water supply in both a fresh and hardened state [36]. Primarily, SAP is a natural or synthetic water-insoluble three-dimensional network of polymeric chains, with the ability to absorb aqueous fluids from the environment [36,37,38,39,40]. Figure 1 shows the swelling process of SAP and the formation of hydrogel when the polymer contacts with water.

SAPs are generally based on polyacrylamide and polyacrylic acid (or their modifications) (Figure 2). The crosslinked three-dimensional structure of SAPs, when in contact with water, enables the formation of a hydrogel almost instantaneously [36]. Different factors such as hydroxides of alkali metals in their composition (usually sodium or potassium) can influence the ability of SAPs to absorb and desorb in different environments by affecting the crosslinking density of SAP.

The most remarkable successes of SAPs have been in the mitigation of autogenous shrinkage [31,37,38,39], plastic shrinkage [36,40], enhancing freezing and thawing resistance [40,41], self-sealing [42,43], and self-healing [44,45] in various types of mortar and concrete. However, SAP effectiveness in cementitious matrices depends not only on its intrinsic characteristics (shape, size, crosslinking density, chemical structure, and composition) but also on the ionic concentration of the surrounding pore solution [36,46,47,48,49,50,51,52,53,54].

Swelling kinetics and the final long-term storage capacities of SAPs are also depended on the ionic strength of the aqueous solution, which may change significantly inter- and intramolecular interactions of polyelectrolytes due to the shielding of charges on the polymer chain [36,46,55,56]. The presence of alkali ions in the polymer chain results in an increase of absorption capacities of SAPs due to a strong interaction between water and alkalis fixed to the polymer chain [57,58]. On the contrary, ions in the pore solution of cementitious mixture can cause additional interlinking of the polymer chains and perform less swelling. The reason behind this is to increase the concentration of ions outside the SAP, which leads to a decrease of the osmotic pressure inside the gel, and consequently to the reduced swelling of SAP [36,48]. For example, the polymer including of hydrophilic groups (-OH, -CONH, -CONH_2_, -COOH) can interact with each other (see Equation (1)) by hydrogen bonding [59].
(1)$$-{\mathrm{CONH}}_{2}\to -\mathrm{COOH}↔-{\mathrm{COO}}^{-}+H^{+}\;$$

This leads to an increase in crosslinking degree and results in a stiffer and more brittle material, i.e., leading to reduced water absorption. One the other hand, SAPs containing potassium or sodium salts may display a better performance. This is because alkali ions weaken the formation of hydrogen bonds between hydrophilic groups and strengthen the crosslinking density, helping to increase water absorption capacity [60,61]. However, SAPs with values above 6.6 wt % for potassium [62] and 10 wt % for sodium [63] may decrease significantly the swelling capacity of the polymer due to the increased ionic concentration of external aqueous solution.

Thus, the main purpose of this paper was to show the differences of SAP sorption characteristics in different filtrate solutions prepared with three commercially available cements. The studied SAPs had different chemical compositions and practical grading. Experimental results of their sorption performance in different solutions, including deionised water (DI), Portland cement (PC), and blended cements (PC-FA and PC-GGBS) were analysed. In addition, the pH of different solutions was monitored over time.

## 2. Materials and Research Methodology

### 2.1. Cements

Three types of cement have been used, including CEM I 52.5N (Portland cement-PC), CEM II/B-V 32.5R (PC-FA), and CEM III/A 42.5N (PC-GGBS). CEM I, II and III have been supplied by Hanson Cements (Stamford, UK), Lafarge (Leics, UK) and Ecocem (Dublin, Ireland) respectively. The chemical and physical characteristics of cements as provided by the manufacturers are presented in Table 1.

The shapes and sizes of cements particles were characterised by scanning electron microscopy (SEM) (EVO 50, Carl Zeiss AG, Oberkochen, German) image analysis and presented in Figure 3. The particle size distributions (PSD) of cements were determined by the Laser Diffractometry (Mastersizer X, Malvern Panalytical, Malvern, UK), and results are shown in Figure 4.

### 2.2. Superabsorbent Polymers (SAPs)

Three types of cross-linked superabsorbent polymers (called here: A, C, and E), provided by the BASF Construction Chemicals GmbH (Trostberg, Germany) were used in this study (Table 2).

SAP A is copolymer of acrylamide and acrylic acid; SAP C, with similar particle sizes to SAP A, comprises a modified polyacrylamide; and SAP E has a similar molecular structure to SAP C (modified polyacrylamide) but different particle grading. The physical and chemical characteristics of SAPs have been evaluated in this study.

### 2.3. Experimental Method

SAP particle size distributions were determined by the laser diffraction method. The shapes and size of SAPs were also characterised by the SEM techniques [49]. Chemical characteristics were analysed by the Energy-dispersive X-ray Spectroscopy (EDX) (Aztec Energy acquisition software with X-MaxN and X-act Silicon Drift Detector, Oxford Instruments NanoAnalysis, Abingdon, United Kingdom) and Raman Spectroscopy (WITec alpha300 R Raman spectrometer, Ulm, Germany) [49], in dry and wet condition (considering 5 min of water contact).

The absorbency and pH testing were carried out in four different solutions environments: deionised water (DI), CEM I solution, and blended cements (CEM II (PC-FA), CEM III (PC-GGBS)) solutions. Cements filtrate solutions were obtained by mixing the selected cement in DI water with a water-to-cement ratio (w/c) of 5 [46]. Cement slurries (cement + water) were stirred for 24 h with a mechanical stirring, and then separated from the filtrate solution using vacuum filtration [46,47,48,49,50].

In particular, the kinetics of absorption and water absorption capacity (WAC) of SAPs have been determined gravimetrically by the tea-bag method [46,47,48,49,50,51,52]. A typical tea-bag with opening mesh below 20 μm has been used (Figure 5a). The amounts of dry SAP particles were approximated to 0.1 g for DI water and to 0.3 g for cementitious filtrates (Figure 5b). Since the absorption of DI water is higher, a smaller amount of SAP has been used; otherwise, the swollen SAP could exceed the maximum capacity of the tea-bag and give rise to restraints that would hinder a free sorption (Figure 5c) [50]. First, a predetermined amount of dry SAP was inserted in a tea-bag that was pre-wetted in the test liquid. Next, the filled tea-bag was hung in a 200 mL beaker, containing the test liquid. The beaker has been tightly covered with a self-adhesive plastic stretch film during the measurement in order to limit carbonation and evaporation. The tea-bags with SAP gel were weighed at 1, 5, 10, 30, 60, and 180 min, and 1, 2, and 3 days after the SAP/liquid contact time. Three individual tea-bags were tested for each SAP sample to ensure reliability of the results. The pH of all solutions was measured during the tests in order to verify the effect of carbonation of cement slurry solution, which could affect the SAP absorption behaviour.

Before weighing, the tea-bag was placed on a dry cloth and gently wiped for a short time (maximum 30 s) in order to remove surplus and weakly bound liquid. It should be noted that the tea-bag should not be squeezed, as this could disturb the degree of sorption. After weighing, the tea-bag containing the SAP gel was returned to test fluid until the next weighing step. In order to account for the wet dead load of each tea-bag in the further procedure, the average mass of an empty wetted tea-bag was given by using at least ten individual tea-bags (Equation (2)).
(2)$$m_{0}=\frac{1}{n}\times \overset{n}{\underset{i=1}{\sum }}\left(m_{Bi}-m_{Ai}\right)\;\;\;\;\;\;n\geq \;10\;\;\;\;\;\;$$
where *m*_0_ is the average wet dead load of the tea-bags; *n* is the number of tea-bags used, *n* ≥ 10; *m_Ai_* is the individual dry mass of each tea-bag; *m_Bi_* is the individual wet mass of each tea-bag.

The water absorption capacity (*WAC*) of SAPs was calculated by Equation (3):(3)$$WAC=\frac{m_{3}-m_{2}-m_{0}}{m_{2}-m_{1}}$$
where *WAC* is the water absorption capacity (g/g); *m*_0_ is the average wet dead load of the tea-bags (g) (Equation (2)); *m*_1_ is the mass of the dry tea-bag (g); *m*_2_ is the mass of the dry SAP-containing tea-bag (g); *m*_3_ is the mass of the swollen gel-containing tea-bag at a specific time of soaking (g). The tea-bag method is a standard test and widely accepted quantification technique for the analysis WAC of SAPs [46,49].

All analyses of SAP samples were carried out in standard laboratory conditions at a temperature of 21 ± 2 °C and a relative humidity of 40 ± 5%.

## 3. Characterisation of SAPs

This section focusses on the physical and chemical characteristics of SAPs, as well as their sorption performance in different solutions, including deionised water (DI), CEM I filtrate, and blended cements (CEM II (PC-FA) and CEM III (PC-GGBS)) filtrates.

### 3.1. Physical Characteristics of SAPs

The average particle sizes and corresponding standard deviations are presented in Table 3 (samples tested in triplicate). Figure 6 shows the average results of particle size distribution (PSD) of SAPs.

As seen in Figure 6, SAP E has notably smaller particle sizes (between 20 and 130 μm) than others. SAP A and SAP C have particles in the same size range, predominantly between 30 and 140 μm. Some small differences of sizes and shapes of particles were observed on the SEM micrographs (Figure 7). The images in Figure 7 also show all SAP particles in wet condition. Some characteristic features of hydrated networks of the swollen gels can be identified.

At low magnification (100 µm), different networks of SAP A, C, and E are apparent. The network’s similarities between SAP C and E result from the same chemical composition. Considering that all SAPs were allowed to swell during the similar period of time (same time of water exposition), SAP A seems to have a faster water absorption in the first 5 min. While SAP A has formed a clear swollen gel, SAP C and E still have shown very well-defined particles. However, at the higher magnifications (Figure 7c,d,g,h,k,l), the effect of network density becomes more apparent, depending on the chemical composition, size, and shape of the polymer particles. SAP C shows a denser and more closed swollen structure than SAP A and E in water. This may reflect on the WAC performance (as further discussed), indicating qualitatively that SAP C has the lowest water absorption capacity.

### 3.2. Chemical Characteristics of SAPs

Figure 8 presents the chemical characterisation of SAP samples with respect to their elemental chemical composition. In particular, alkali contents (especially Na^+^ and K^+^) were analysed as they significantly influence the progress of absorption and potential individual desorption of SAPs [36]. Similar elementary compositions of SAP C and E, regarded to the primary molecular structure, are clearly evidenced (with the exception of sulphur content). The presence of sulphur (in the form of S^2−^) in SAP can contribute to the greater stability due to the formation of disulphide bonds. These are very strong bonds that can hold polymers in their respective conformations and, therefore, they play an important role in their folding and stability [64]. Disulphide bonds can only be cleaved by external stimulus, such as oxidation-reduction potential [65]. However, Almeida et al. [47] showed that alkali contents have more influence on SAP performance than the presence of sulphur species itself. Figure 8 also shows different types and concentrations of alkali ions in different SAPs. The alkali contents (Na^+^ + K^+^) were found to be 12.3, 4.0, and 3.8 respectively for SAP A, SAP C, and SAP E. Polymers with ionic groups are formed by cross-linked polyelectrolytes, which dissociate in aqueous solutions. In these polyelectrolytes, the charged molecular chains play an important role in the physical features of hydrogels including structure, stability, and interactions of various molecular assemblies [66].

Generally, SAPs with higher absorption capacity possess a lower cross-linking density when compared to other types of hydrogels [54]. The results of Raman spectroscopy, as shown in Figure 9, confirm that all SAPs have a polyacrylamide structure base. Again, a very similar molecular structure of SAP C and SAP E can be identified, by comparison of the peaks position.

### 3.3. pH of SAP Solutions

Figure 10a shows the results of pH analysis of SAPs in DI water over the period of 7 days. Very similar, almost linear behaviour has been recorded for all SAPs. SAP A, SAP C, and SAP E had the pH values about 7.2, 6.8, and 7.4, respectively. Small divergences observed in pH for SAP solutions may be related to the interaction and stabilisation of the polymer base given by differences in cross-linking dissolutions in deionised water [54]. The total alkali content itself seems to be of a minor significance because SAP A (with the highest alkali content) had a similar pH to SAP E. K^+^ and Na^+^ ions can be strongly adhered to the polymeric chain, which are not easily dissolved in water, interfering in pH measurements.

Figure 10b shows pH analysis of CEM I (PC), CEM II (PC-FA), and CEM III (PC-GGBS) filtrate solutions as reference solutions over the period of 7 days. The CEM II solution containing FA had the lowest pH due to a lower alkalis concentration, which is usually found in SCM systems [35]. Similar comments can be made for CEM III (PC-GGBS), which had lower pH than CEM I (PC) filtrate solution. The differences between the pH level of each cementitious solution can be mainly related to their respective calcium concentration: CEM I (highest pH), CEM III, and CEM II (lowest pH) had, respectively, 64.3%, 57.1%, and 43.5% of CaO content (Table 1). Thus, this pH variation is associated with decreased alkali contents (mainly by calcium contents) and hydroxide concentrations, due to the reactions of SCMs. Additionally, in the case of FA, the reaction of pozzolanic materials with Portlandite leads to lower pH values of pore solution [65]. In case of PC-GGBS, this variation could be related to the formation of reduced sulphur species (HS^−^, SO_3_^2−^, and S_2_O_3_^2−^) [34,67]. Moreover, it was found that in all cementitious solutions, pH dropped considerably after the second day, as a result of carbonation. Carbon dioxide from the atmosphere can react with cement filtrates (mainly Ca(OH)_2_), forming basically CaCO_3_ and reducing the pH of the aqueous solution [68].

Figure 11a–c show the results of pH measurements of SAP solutions in CEM I (PC), CEM II (PC-FA), and CEM III (PC-GGBS) filtrate solutions. It was found that the SAPs in CEM I solution had a similar pH to their respective reference solution over time. Furthermore, in blended cements solutions, all the results for SAPs were similar, up to the second day. However, after this period, they behaved differently depending on the type of polymer but had a similar trend regardless of the type of SCM.

Overall, all SAPs had similar pH values in the same cementitious solution up to the second day. This convergence in early ages can be attributed to the complex formation between the anionic functional groups of the polymer and Ca^2+^ from cements, which decrease the efficient charge density of anionic groups [46,50,54]. Moreover, this tendency was similar for both blended cements solutions: all SAPs displayed the same behaviour in CEM II (PC-FA) and CEM III (PC-GGBS) filtrates.

After the second day, SAP performances were very dependent on the type of cements. Although SAP samples indicated comparable pH behaviours in blended cements solutions, pH values of all SAPs exhibited a stable performance in CEM I (PC) filtrate solution. These results showed that not only the SCMs can affect pH values of cementitious solutions but also that the studied SAPs contribute to a significant change in these solutions, especially when carbonation takes place.

At the second day, all tested SAPs solutions showed higher pH values compared to that of pure CEM II (PC-FA) and CEM III (PC-GGBS) filtrates. This may be at least partially caused by chemical interactions of alkali in the polymers network. Alkali metals tend to lose one electron and form ions with a single positive charge (e.g., Na^+^ and K^+^). Once dissolved in solution, they can easily react with water to produce hydroxides, increasing the pH of cementitious solutions. This high pH sensitivity can be related to the changes in cementitious concentrations between the total alkalis from polymer composition and the solution’s compounds [54].

However, this effect mainly depends on the type of polymeric base, and the total alkali content itself seems to be of lesser importance. This is because SAP A (with the highest total alkali content) had an intermediate pH value after the second day. Thus, the molecular structure of SAP E may have a lower ability to retain its positively charged ions in the polymeric chain, making this SAP more vulnerable and susceptible to lose alkalis in cementitious aqueous solutions (especially with SCM). This was reflected in higher pH values after the second day in CEM II and CEM III solutions.

On the contrary, SAP C, with similar chemical composition, showed lower pH values than SAP E. However, the similar trend was observed for pure Portland cement, indicating that this polymer is the most stable in all cementitious solutions. This can be also related to the considerable content of sulphur elements in SAP C (Figure 8), which can lead to the formation of disulphide bonds [47]. These are very strong bonds that only can be cleaved by external stimulus, such as oxidation-reduction potential [69].

Finally (at day-7), pH values around 10 were recorded for CEM I (PC) and CEM III (PC-GGBS) systems and 9.5 was recorded for the CEM II (PC-FA) system, regardless of SAP type.

### 3.4. Absorption Capacity and Kinetics of SAPs

The sorption capacity of SAP represents the long-term equilibrium amount of absorbed and desorbed liquid. In turn, the maximum absorption capacity indicates the greatest value of water intake by the polymer, regardless of the time when it occurs. Moreover, the sorption kinetics of SAP is described as their progressive ability to absorb and swell when in contact with an aqueous fluid by forming a hydrogel and potentially desorbing it [52].

Figure 12a,b illustrate the results of the analysis of sorption behaviour of SAPs in DI water by the tea-bag method during the first 180 min and 1440 min.

As shown in Figure 12a, an initial intensified swelling was observed during the first 30 min. At 180 min, the values of water absorption capacity (WAC) were around 252, 260, and 286 g/g, for SAPs A, C, and E respectively. Figure 12b illustrates the rate of water absorption during a longer period of time, and maximum values of WAC were found at the end of the test. Increasing absorptivity trends were exhibited by all SAPs up to 40%, 10%, and 30%, respectively for SAPs A, C, and E at 24 h immersed in DI water.

Figure 13a,b show the WAC values in CEM I (PC) solution. Overall, the cementitious solutions had a noticeably reduced swelling capacity of polymers (around 10× lower than in DI water). In CEM I (PC) solution, all SAPs started to release water after a longer period of time (up to 3 h). Although SAPs C and E seem to be slightly unchanged over time, overall, the WAC of SAP A was markedly reduced. The max WAC in CEM I (PC) solutions was 38, 40, and 42, respectively for SAPs A, C, and E (measured at 180 min). However, these were significantly reduced at 24 h to 34, 38 and 42 g/g, for SAPs A, C, and E, respectively.

Figure 14a,b present the results of SAPs absorbency in CEM II (PC-FA) solutions. Similar behaviour of SAP A and SAP C absorption was recorded compared to CEM I solution. WACs in CEM II (PC-FA) filtrate at 24 h were about 33, 37, and 46 g/g correspondingly for polymers A, C, and E. However, the max WACs registered were 37, 38, and 46, for SAPs A, C, and E, respectively.

Figure 15a,b illustrate absorption curves of SAPs in CEM III (PC-GGBS) solution. Similar behaviour (to CEM I) of SAP absorption was recorded again: SAP E had the ability to maintain most of the initial max WAC, while the WAC of SAP A was significantly reduced over the time (SAP C had a very little water release). Furthermore, a considerable alteration in water storage and release ability was recorded. Final WAC values (at 24 h) reached approximately 25, 33, and 40 g/g, respectively for SAP A, C, and E.

Substantial reductions were found in the swelling capacity of SAPs in cements solutions (blended or not) when compared to DI water (Figure 12, Figure 13, Figure 14 and Figure 15). This could be attributed to the presence of dissolved ions in binder filtrates, especially K^+^, Na^+^, Mg^2+^, and Ca^2+^, which reduce the osmotic pressure and restrict swelling [50,54]. SAP samples displayed the highest absorption capacity in CEM II (PC-FA) solutions, and the lowest swelling capacities were found in CEM III (PC-GGBS) solution. In particular, SAP E had the greatest overall WAC value in CEM II (PC-FA) system.

In all cement filtrate solutions (see Figure 13, Figure 14 and Figure 15), SAPs displayed intensified absorption during the first 30 min, reaching their maximum. However, after that time, the SAPs absorbency curves shifted as a function depending on the polymer type. Different SAPs, depending on individual monomers, may have distinct behaviour in contact with cementitious solutions. However, different cementitious solutions lead to a similar trend of absorption capacity considering individual SAPs, indicating that each polymer may have an expected comparing performance in those solutions. It is important to stress that the overall trend should be interpreted in a qualitative rather than quantitative way. For example, in all cementitious solutions (PC, PC-FA, and PC-GGBS), SAP E had the highest WAC at 24 h, followed by SAP C and then SAP A, although individual results were different in each system.

Therefore, the obtained results of swelling capacities of SAPs in cements solutions show that SAP E has the highest capacity (42–46 g/g), regardless of the cement solution. In fact, SAP E was the polymer with the least dense and more open swollen structure (in wet condition) observed in Figure 7. Furthermore, SAP A released a smaller portion of absorbed cement filtrate solution, featuring a WAC of 25–33 g/g at 24 h. A more stable swelling was recorded for SAPs C and E (in PC solution) and showed low or no desorption over time (in PC-FA and PC-GGBS solutions). These results can be related to several factors including molecular structure, cross-linking density, the grading and anion concentration of SAPs, and their interactions with cement filtrate solution [46,54], as discussed below.

All SAP samples exhibited an initial swelling and increasing absorptivity trend in deionised water (up to 30 min, Figure 12a). The stable swelling behaviour in DI water was recorded for SAP C after 24 h of exposure time (Figure 16a). SAP A and SAP E reached a maximum of their absorption capacity after about one day of testing, which was followed by a similar and gradual increase.

The difference observed in their swelling capacity was due to the individual polymer base structure and different alkalis contents [62]. This ability to absorb more water is a result of a higher potassium content of SAP A (Figure 8) and smaller particle sizes of SAP E (Figure 6), which led to increased polymer network volume and increased specific surface area, respectively [54]. Specifically, to SAPs with greater surface area, more water can be trapped between the hydrogel particles, leading to increased overall WAC measured by the tea-bag method.

Extensive reductions in overall swelling capacities of SAPs (compared to DI water, Figure 16a) were observed in cements solutions. The comparison of absorption capacity of SAP A in different cements solutions is shown in Figure 16b for values obtained for 3 days of testing.

Overall, SAP A had the lowest absorption capacity in CEM III (PC-GGBS) solution (24 g/g), and in CEM I (PC) and CEM II (PC-FA) solution, this polymer had similar maximum absorption values: 38 and 36 g/g, respectively, with standard deviation lower than 1 g/g. This was reached after about 10–30 min of contact with the fluids, which was followed by a decrease or release of water. This indicates losses of fluid retention capacity, where this desorption of SAP A was much more pronounced in CEM III (PC-GGBS) solution.

A more stable swelling behaviour in cements filtrate solutions was observed for SAP C (Figure 16c), especially for CEM I and II filtrates. The maximum absorption capacity was recorded after about 30 min of testing, with values reaching around 37, 38, and 36 g/g in CEM I, CEM II, and CEM III solutions, respectively. SAP C behaves similarly in CEM I (PC) and CEM II (PC-FA) solutions, and the lowest water absorption was recorded in CEM III (PC-GGBS) solutions during the test. Although SAP C displayed a slight decrease of absorbency in PC-GGBS solution after 24 h of testing (around 1.1% of reduction), it stayed nearly unaltered over the time in the other systems. This indicates the greater stability of this polymer in cementitious solutions, which may be related to its ideal alkali content, ability to form disulphide bonds, and the potential influence of its molecular structure.

SAP E had the highest capacity to store the same amount of water for longer periods, regardless of the type of cementitious material used (Figure 16d). SAP E had the highest absorption capacity in CEM II solution, reaching a predominant initial value of 44 g/g (with standard deviation lower than 1 g/g) around 10–30 min. After this time, the WAC of polymer kept a considerable upward trend, showing a value of 48 g/g at 3 days of testing. Even though the long-term equilibrium was not achieved, this measurement was adopted as the max WAC in CEM II solution, since it is comparable (by the time of water exposition) to other SAPs studied. SAP E exhibited a similar absorptivity trend in both CEM I and CEM III. When the maximum absorption capacities were reached after about 10–30 min of contact with the fluid, a slight water release took place. However, this desorption (at the level of 1% approximately) was shortly recovered; after 24 h, SAP E showed a gradual swelling in PC and PC-GGBS.

Thus, it seems that the type of polymer, in terms of physical and chemical composition, is the main factor influencing the sorption characteristics of SAP E in different cementitious systems. The smallest particle sizes (Figure 6) and the lowest amount of K^+^ (Figure 8) increase the osmotic pressure difference between the polymer and cementitious aqueous solutions, which results in a faster water absorbency [54,65]. Furthermore, diffusional processes are involved the swelling behaviour of SAPs, which release water from the SAP particles [58].

Overall, a drop in the absorption capacity of SAP A, a steady swelling behaviour of SAP C, and the largest absorption of SAP E have taken place in all the cement solutions. However, the unstable behaviours of SAP A (after the first hour), SAP C, and SAP E (at 24 h) were more pronounced in the CEM III (PC-GGBS) system (although SAP A had considerable desorption for all cementitious solutions). This effect was may be explained as polymer degradation [51,54,70,71], which decrees network integrity. It can be ascribed to the higher cation concentrations in solutions, resulting in concomitant reduced hydrolysis from the polymer [72,73]. The highest trends with respect to the desorption behaviour of SAPs were observed in PC-GGBS systems (with reduced Ca^2+^ content, as shown in Table 1). This indicates the greater sensitivity of these SAPs on this solution. On the contrary, as expected, the final long-term storage capacities in PC and PC-FA systems were clearly higher for SAPs when compared to PC-GGBS (see Figure 16b–d).

The results of pH analysis also showed the instability of SAPs in blended cements systems (see Figure 11). This can be explained by its high alkalinity (Figure 11b,c). The screening effect of monovalent cations K^+^ and Na^+^ in the SAPs’ chains reduces the hydrogen bonds formation and forms complex bonds between the polymeric network and metal ions from cementitious solutions. Figure 17 shows a schematic overview of the chemical structure of SAPs in DI water, PC solution, and blended cements solutions.

The presence of S^2−^ ion in SAP C and E could also contribute to the greater stability of the polymer due to the formation of disulphide bonds [47], as observed in Figure 16c,d. This ion can form strong bonds, which are able to hold polymers in their respective conformations [64], avoiding polymer degradation.

In addition, high alkali content (especially from SAP A) and its reaction with –COO- groups can lead to degradations of the polymers in blended cements systems. The effect of various ions including K^+^, Na^+^, OH^−^, and Ca^2+^ released into pore solution during hydration [73], on SAP particles leads to different swelling kinetics. The diffusion of monovalent cations such as K^+^ and Na^+^ leads to interaction with a cross-linked potassium/sodium hydrophilic network.

The divalent cations (Mg^2+^ and Ca^2+^) have been found to have stronger interactions with SAP than monovalent ions [75]. However, [50,76] reported that the presence of Ca^2+^ ions in the aqueous fluid decreases the initial swelling of SAP due to bonding with carboxylate groups. However, the swelling gradually recovers as a result of increase of the crosslinking degree during the prolonged contact time.

The contact of trivalent cations (Al^3+^ and Fe^3+^) with the SAP particle slowly establishes cross-links [58] and significantly decreases the swelling capacity of SAP [77]. Apart from this, the sorptivity of SAPs in different cements solutions is affected by the presence of cations. For instance, the instability and greater sensitivity of SAP A in the CEM III (PC-GGBS) system is more likely due to the possibility of interaction of the potassium/sodium from SAP with Mg^2+^; there is more Ca^2+^ from cementitious solutions, since Mg^2+^ is 2.4 times higher in GGBS than PC (Table 1). The highest absorption capacity of SAPs is also more evident in CEM II (PC-FA). This result may be explained by the lower amount of divalent cations in this solution (Table 1). The diffusion of divalent cations in blended cement solutions can form strong complexes with the polymer chain and consequently increase the crosslink density of the polymeric network [50,54,78] and reduce swelling capacity in time [46,48,50]. Further reasons can include ability to form the complex bonding of the acrylate groups by divalent cations including intermolecular and intramolecular complexes, or the presence of one multivalent ion in the solution can neutralise several charges inside the gel [78]. The chemical interaction of trivalent cations from blended cements can also lead to the formation of complex bonds with those SAPs. CEM II (PC-FA) had the highest Al_2_O_3_ content (13.13%), followed by CEM III (8.99%) and then CEM I (4.99%) (Table 1). This is most likely related to this trivalent ion diffusing into the system more slowly than Ca^2+^, thus allowing the sample to reach a greater swelling ratio [77].

## 4. Conclusions

Based on the obtained experimental results, the following observations and conclusions can be drawn:CEM II (PC-FA) solution had the lowest pH due to lower alkalis concentration. Similar comments can be made for CEM III (PC-GGBS), which had lower pH than CEM I (PC) filtrate solution. This is due to the decreased alkali contents and hydroxide concentrations, which in turn are related to the pozzolanic reaction of FA; the pozzolanic reaction with portlandite leads to lower pH values of the pore solution. In the case of PC-GGBS, lower pH values could be associated with the formation of reduced sulphur species (HS^−^, SO_3_^2−^ and S_2_O_3_^2−^). It was found that in all cementitious solutions, pH dropped considerably after the second day, which was most likely as a result of carbonation;SAPs in CEM I (PC) solution had a similar pH to their respective reference solution over time. However, in blended cements solutions, all SAPs displayed different behaviour depending on the type of polymer. This is the result of the complex formation between the anionic functional groups of the polymer and Ca^2+^, which decrease the efficient charge density of anionic groups. Overall, SAP E with a smaller particle size had the highest pH values, followed by SAP A and SAP C;Considerable reductions in swelling capacity were found for SAPs in cements solutions when compared to DI water. This could be attributed to the presence of dissolved ions in binder filtrates, especially K^+^, Na^+^, Mg^2+^, and Ca^2+^, which reduced the osmotic pressure and restricted swelling;SAP samples exhibited the highest absorption capacities in CEM II (PC-FA) solutions, and the lowest swelling capacities were found in CEM III (PC-GGBS) solutions. In particular, SAP E had the greatest WAC value for a PC-FA system due to smaller particle sizes, which resulted in an increased polymer network volume and specific surface area. Moreover, the lowest amount of K^+^ in SAP E increases osmotic pressure difference between the polymer and the cementitious aqueous solutions, affecting the diffusional processes involving the swelling behaviour of SAP;More stable swelling behaviour in cements filtrate solutions were observed for SAP C (specially for CEM I and CEM II systems). It stayed nearly unaltered over the time in both cements solutions, which may be related to its ideal alkali content, ability to form disulphide bonds, and potential influence of its molecular structure;The lower amount of divalent cations in CEM II solution compared to CEM III and CEM I led to the highest absorption capacities of SAPs. In blended cement solutions, these cations can form strong complexes with the polymer chain and consequently increase the crosslink density of the polymeric network. The chemical interaction of trivalent cations can also lead to the formation of complex bonds with those SAPs in blended cement environments, especially in CEM II (PC-FA), since amount of Al^3+^ concentration is higher than that of CEM I (PC).

## Figures and Tables

**Figure 1 materials-14-01609-f001:**
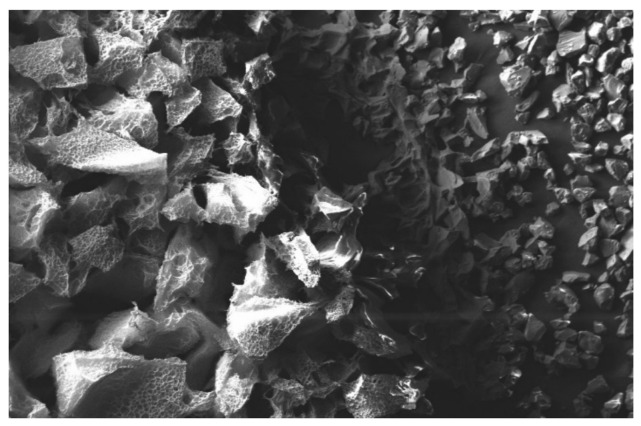
SEM micrographs of superabsorbent polymers (SAP) in dry and wet conditions (100 µm): dry particles (**right**) and swollen hydrogel (**left**) [8].

**Figure 2 materials-14-01609-f002:**
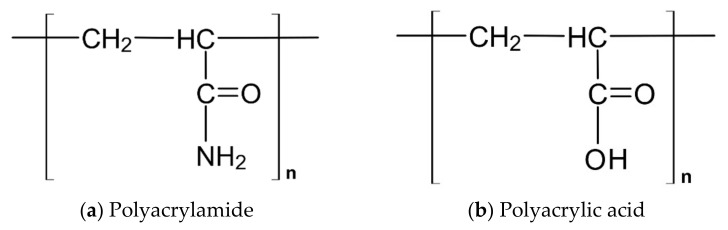
Base of superabsorbent polymers: acrylamide (**a**) and acrylic acid (**b**).

**Figure 3 materials-14-01609-f003:**
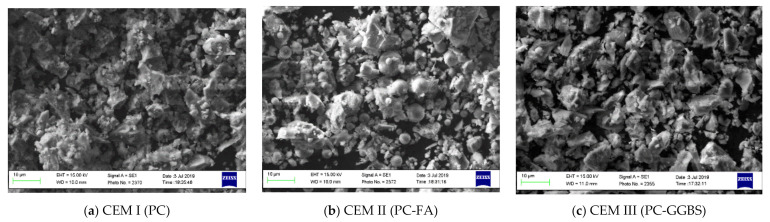
SEM micrographs of CEM I (PC) (**a**), CEM II (PC-FA) (**b**), and CEM III (PC-GGBS) (**c**). PC: Portland cement, FA: fly ash, GGBS: ground granulated blast-furnace slag.

**Figure 4 materials-14-01609-f004:**
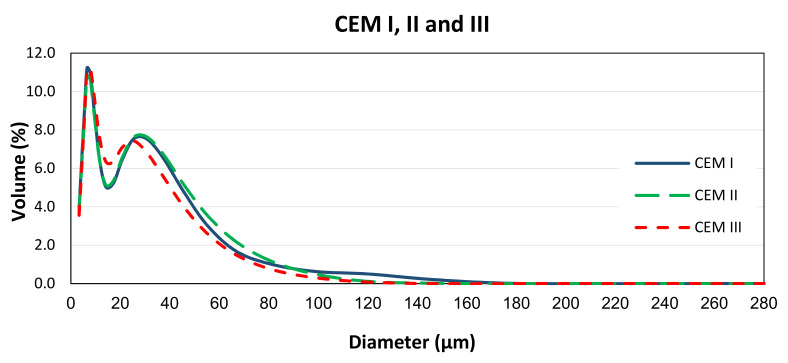
Particle size distributions of CEM I (PC), CEM II (PC-FA), and CEM III (PC-GGBS).

**Figure 5 materials-14-01609-f005:**
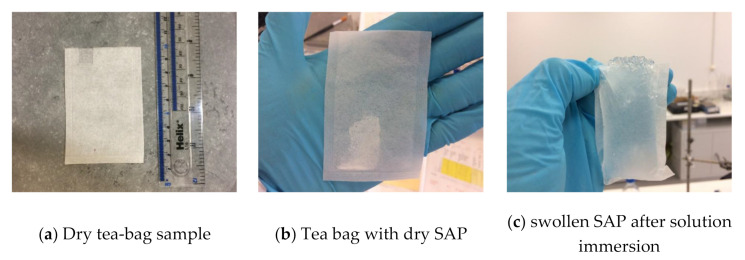
Tea-bag test (**a**–**c**).

**Figure 6 materials-14-01609-f006:**
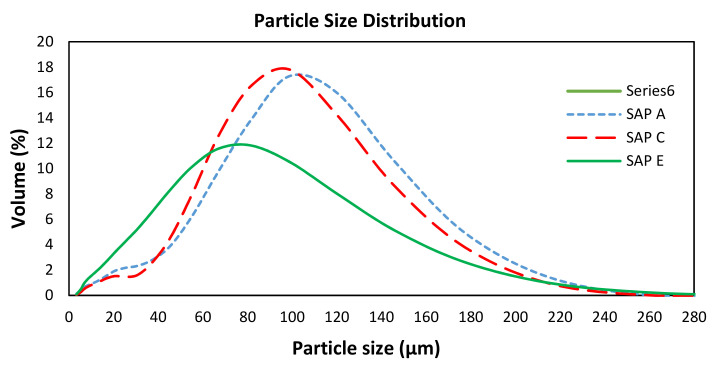
Particle size distribution of SAPs [31].

**Figure 7 materials-14-01609-f007:**
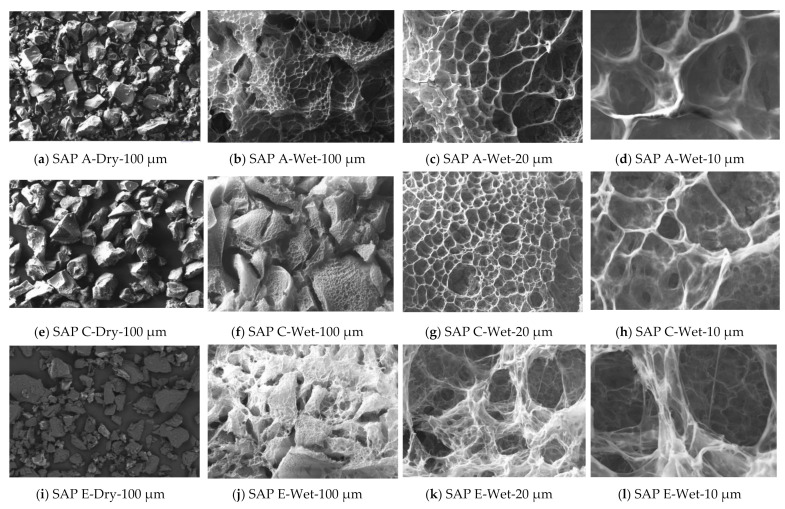
SEM of SAPs in dry and wet conditions; SAP A (**a**–**d**), SAP C (**e**–**h**), and SAP E (**i**–**l**).

**Figure 8 materials-14-01609-f008:**
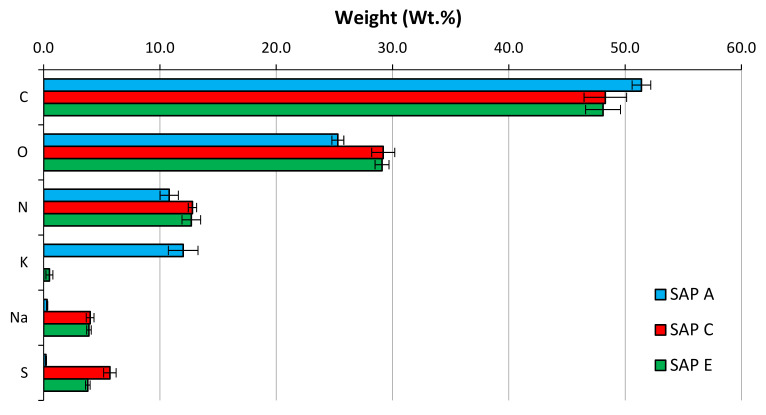
Chemical characterisation of SAP samples.

**Figure 9 materials-14-01609-f009:**
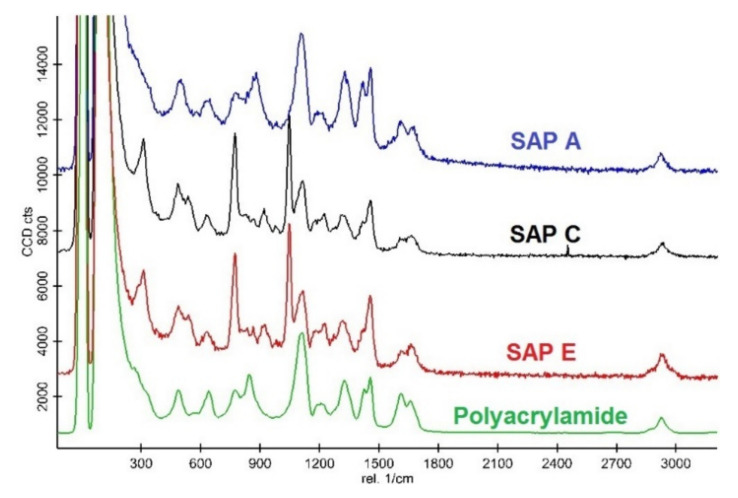
Molecular structure characterisation of SAP samples [6].

**Figure 10 materials-14-01609-f010:**
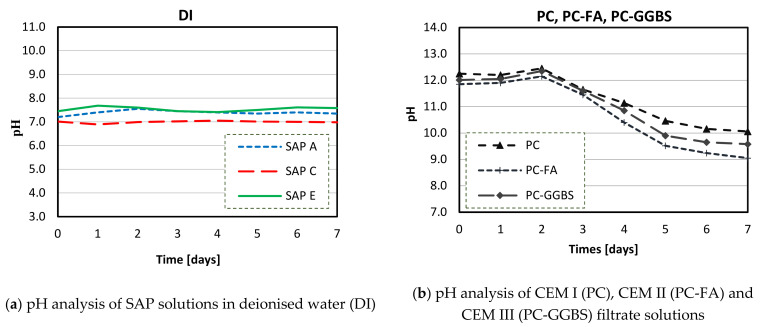
pH analysis of SAP in deionised (DI) water (**a**) and pH analysis of cements filtrate solutions (**b**).

**Figure 11 materials-14-01609-f011:**
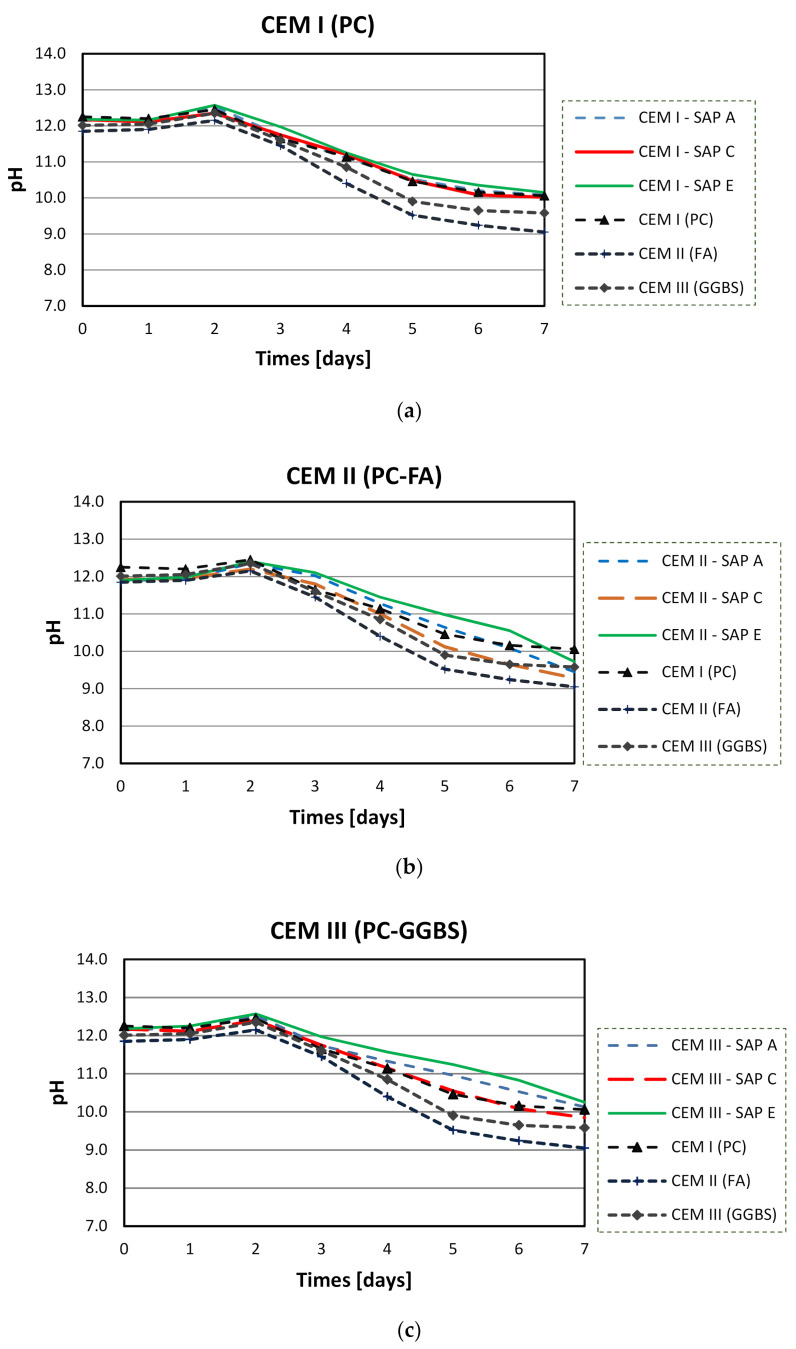
pH analysis of SAP solutions in CEM I (PC) (**a**), CEM II (PC-FA) (**b**), CEM III (PC-GGBS) (**c**) filtrates.

**Figure 12 materials-14-01609-f012:**
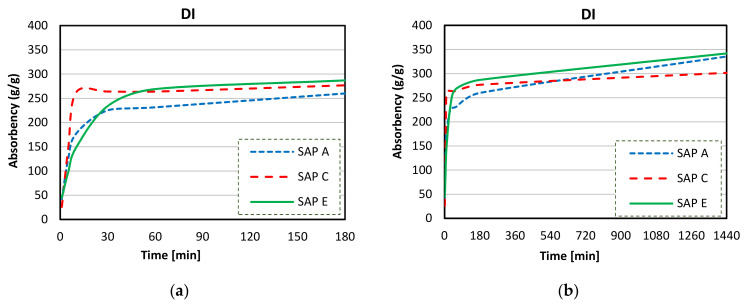
Sorption behaviour of SAPs in DI water up to 180 min (**a**) and up to 1440 min (**b**).

**Figure 13 materials-14-01609-f013:**
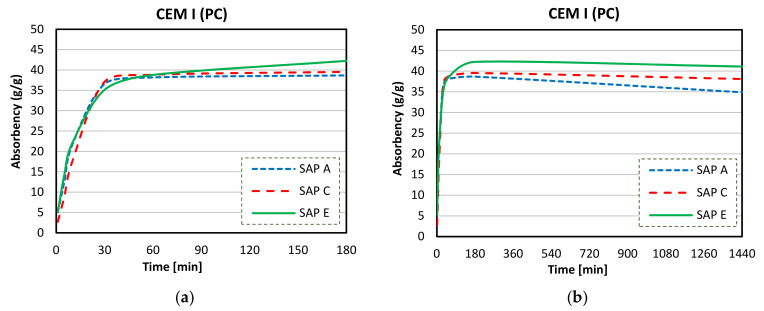
Sorption behaviour SAPs in CEM I (PC) up to 180 min (**a**) and up to 1440 min (**b**) [6].

**Figure 14 materials-14-01609-f014:**
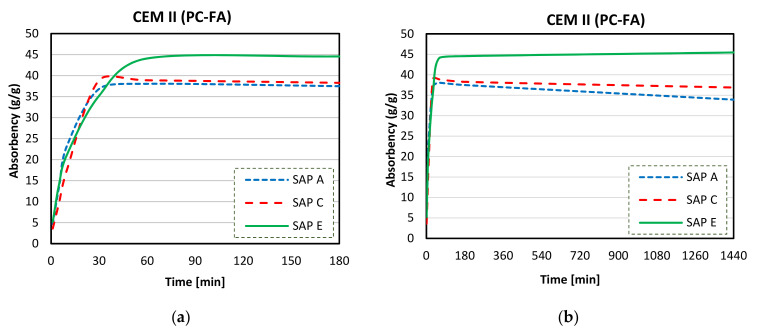
Sorption behaviour of SAPs in CEM II (PC-FA) up to 180 min (**a**) and up to 1440 min (**b**).

**Figure 15 materials-14-01609-f015:**
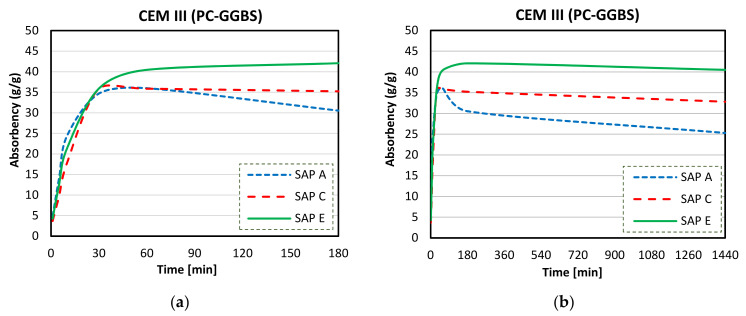
Sorption behaviour of SAPs in CEM III (PC-GGBS) up to 180 min (**a**) and up to 1440 min (**b**).

**Figure 16 materials-14-01609-f016:**
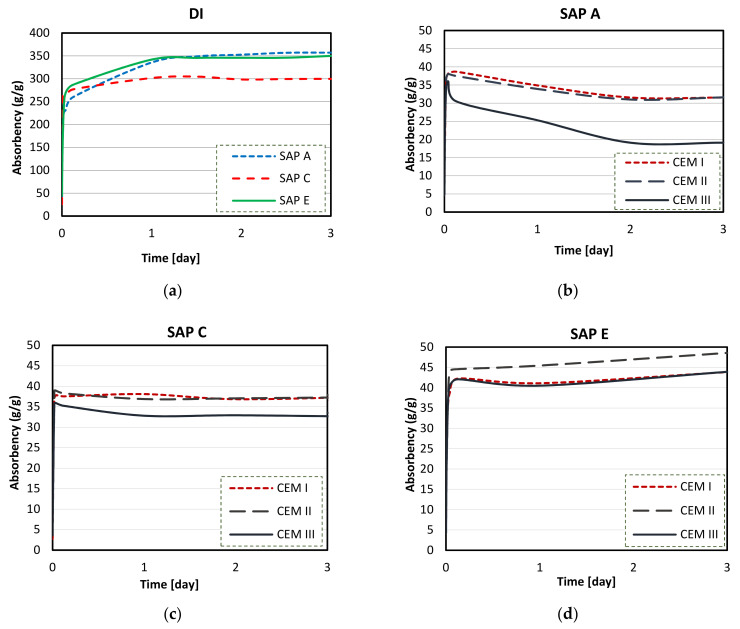
Sorption behaviour of SAPs in different solutions up to 3 days: (**a**) DI water, (**b**) CEM I (PC), (**c**) CEM II (PC-FA), and (**d**) CEM III (PC-GGBS).

**Figure 17 materials-14-01609-f017:**
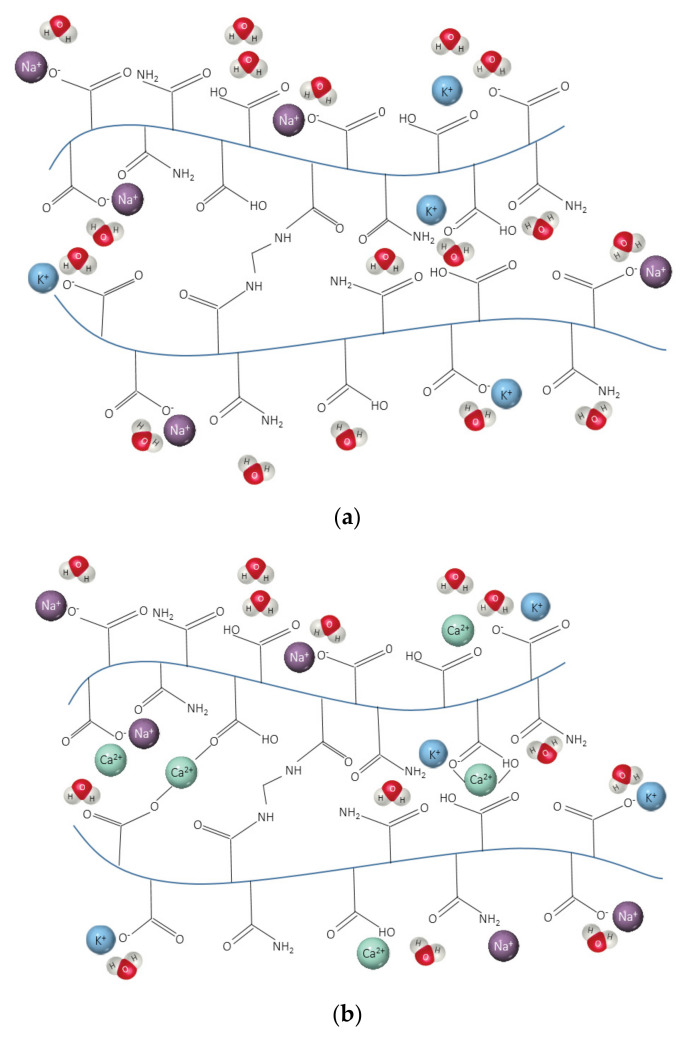
Schematic overview of the chemical structure of SAP in (**a**) DI water, (**b**) in CEM I (PC) solution, and (**c**) in blended cements solutions (adapted from [74]).

**Table 1 materials-14-01609-t001:** Chemical analysis of CEM I, II, and III [6].

Compound (%)	CEM I (PC)	CEM II/B-V (PC-FA)	CEM III/A (PC-GGBS)
SCM (%)	0	30	50
CaO	64.3	43.48	57.13
SiO_2_	20.76	32.69	24.50
Al_2_O_3_	4.99	13.13	8.99
MgO	2.19	1.33	5.33
Fe_2_O_3_	2.57	3.29	1.76
K_2_O	0.27	1.26	-
Cl	0.06	-	0.04
MnO	-	0.07	5.33
TiO_2_	-	0.56	0.58
ZnO	-	0.02	-
Mn_3_O_4_	-	-	0.16
Loss on ignition	2.39	0.16	1.19

**Table 2 materials-14-01609-t002:** General characteristics of studied superabsorbent polymers (SAPs).

SAP Type	Elementary Molecular Structure	Particle Grading
SAP A	Copolymer of acrylamide and acrylic acid	Coarser
SAP C	Modified polyacrylamide	coarser
SAP E	Modified polyacrylamide	finer

**Table 3 materials-14-01609-t003:** Particles size features of SAPs.

Equivalent Diameter	SAP A		SAP C			SAP E
	Average (µm)	SD (µm)	Average (µm)	SD (µm)	Average (µm)	SD (µm)
d(v, 0.5) ^1^	89.55	0.36	84.88	0.53	61.45	0.66
d(v, 0.1) ^2^	26.01	0.67	32.96	1.59	17.76	0.13
Mode ^3^	102.51	0.43	95.19	0.44	76.72	0.22
d(v, 0.9) ^4^	147.24	0.50	140.00	0.76	127.71	1.81

^1^ d(v, 0.5), (volume median diameter) is shown that 50% of the distribution is larger and 50% is smaller than this value; ^2^ d(v, 0.5), d(v, 0.1), it is indicated that 10% of the volume distribution is smaller than this value; ^3^ d(v, 0.5), mode, it means the value that appears most often in a set of data (peak of the curves); ^4^ d(v, 0.5), d(v, 0.9), it is shown that 90% of the volume distribution is smaller than this value.

## Data Availability

Data sharing is not applicable for this article.

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
