# Peer review of "The Effect of SCMs in Blended Cements on Sorption Characteristics of Superabsorbent Polymers"

_materials, 2021, doi:10.3390/ma14071609_

Round 1
Reviewer 1 Report
This paper presents SAP interaction with filtrate solutions from PC and blended cements. The topic is very interesting and important to use SAP in blended cements. The paper can be accept to publish in the journal after the following modification.
- To understand the interaction of SAP with blended cements, pore solution extraction and its concentration are necessary because the filtrate solutions do not represent the pore solution. For example, the pH of the filtrate solution is lower than pore solution.
- Authors should provide the ionic composition of filtrate solution as it varies with addition of slag or fly ash. The composition of ions significantly influence its interaction with SAPs.
- Line 55-56, this is wrong. The microstructure and porosity become denser with SCMs addition.
- Figure 8, please explain the effect of sulfur. The chemical characteristics of SAPs are due to Na+ + K+ or only K+. Some places authors used total alkali content to explain the phenomenon, but in other places only K+. Please discuss appropriately.
- Figure 9. What is green color.
- Explanation for Figure 10 a is wrong because SAP A has high alkali content.
- Authors explanation to link pH and alkali contents are misleading. Please check it throughout the paper, including line 269-275, 289-291.
- How does high K+ SAP absorb more water than others?
Reviewer 2 Report
Dear authors,
Thank you for preparing such a comprehensive and extensive article.
The article presents cements with supplementary cementitious materials in combination with superabsorbent polymers.
I appreciate the combination of three types of cement with three SAPs.
The number of methods that have been used to determine SAP properties is above average. Nevertheless, they are described concisely, clearly and the results are evaluated clearly. In addition, they are logically justified in connection with the need to control hydration in concrete.
I appreciate the scope of the literature.
I have a few questions:
line 190 - from how many samples or measurements were standard deviations obtained? It is not clear to me.
line 144 - I understand well that the tests were performed on samples mixed from cements and SAP?
Is it possible to take into account the effect of aggregate (ph, chemical composition, etc.)? Because it is up to 70% of the content of UHPC.
If it is commented in the article, please help where.
Do the authors evaluate the effect of the superplasticizer, which is almost always used today and could disrupt the benefits of SAP?
Thank you.
A few technical comments:
page 2, figure 2 - empty space looks strange,
page 13, figure 12 - both pictures should be the same size
page 14, figure 14 - both pictures should be the same size
line 533 - please delete "Please add"
Best regards
